# Research Regarding the Coupling and Coordination Relationship between New Urbanization and Ecosystem Services in Nanchang

**Yangcheng Hu \*, Yi Liu \* and Zhongyue Yan**

School of Business Administration, Nanchang Institute of Technology, Nanchang 330099, China
* Correspondence: hycnit@nit.edu.cn (Y.H.); 2020351001@nit.edu.cn (Y.L.)

**Abstract:** The new urbanization (NU) will lead to changed land use types, affecting the ecological environment and ecosystem service value (ESV). The NU is affected by the ecological environment because of resource scarcity when the ecological environment is damaged. NU levels and ESV were used to evaluate the degree of urbanization development and value provided by ecosystem services (ES), respectively, and to analyze their coupling and coordination relationships. This study shows that (1) the Nanchang city NU increases annually, at an accelerated rate, while the city scale continuously expands. Among the various NU subsystems, economy and spatial urbanization are primary, and the remaining subsystems are secondary. (2) In terms of the area of each land use in Nanchang, arable land is the most widely distributed, followed by forest land, and water. The land type with the greatest change was development land, followed by arable land. (3) ESV declined during the study period, with water and forest land being the main ESV components. Hydrological regulation had the greatest contribution among the individual services, while maintaining the nutrient cycle had the minimal contribution. The high-value areas of Nanchang ecology were mainly located in the northeast corner and the water location in the southeast, while the low-value areas were mainly located in the central Nanchang county area. (4) The coupling degree (CD) of Nanchang's NU and ES showed an inverted U-shaped development trend, first increasing and then decreasing. The coupling coordination degree also showed the same trend and exhibited fluctuation in the evolution process.

**Keywords:** ecosystem services; ecosystem service value; land use; new urbanization; coupling and coordination degree

## 1. Introduction

Today, as economic globalization is gradually becoming the general trend of human social development, new urbanization (NU) is seen as an overall indicator of the degree of social development and urban construction level of cities [1,2]. NU has attracted widespread attention and research [3–5]. Since its reform and opening up, China's NU has developed rapidly, but there are some ecological and environmental problems in the development process [6,7]. These include increased air pollution [8], serious water pollution [9], and water shortages [10]. In addition, the development of NU will inevitably change land use and its spatial pattern, which in turn will affect the ecosystem service value (ESV) in the region [11,12]. In 2016, General Secretary Xi Jinping indicated during his inspection in Jiangxi that the green ecology of Jiangxi is its greatest wealth, advantage, and brand. Moreover, Jiangxi must satisfactorily protect the mountains and water, taking a path in which economic development and the improvement of ecological civilization complement each other creating a "Jiangxi model" of beautiful China. The construction of an ecological Nanchang, that combines NU promotion and ecological civilization construction, is not only important to enhance the leading role of Nanchang in the high-quality development of Jiangxi, but also would provide a reference for the high-quality development of other cities

in China. Therefore, it is necessary to study the NU and ecosystem service (ES) changes in Nanchang and its coupling coordination degree (CCD).

New-type urbanization is an important driving force for the sustainable and healthy development of the national economy. The inevitable trend in China is for industrialization and modernization. The current research regarding NU can be divided into theoretical and empirical research. In terms of theoretical research [13–17], the main focus is on the definition of NU. In the early stages of research, urbanization mainly refers to the process of continuous transformation of the agricultural population into a non-agricultural population. As research progresses, scholars find that the traditional concept of urbanization is increasingly unsuited to current development; consequently, the concept of NU is proposed based on traditional urbanization. From current research, NU evaluation methods can be divided into the single-indicator assessment method and compound-indicator assessment method. A single indicator uses one main and most representative indicator to measure NU. The most commonly used indicators are the proportion of the urban population, proportion of the non-farm population [18], or the night light index [19,20]. The composite index assessment method is a comprehensive index for evaluating urban social development and construction from multiple dimensions. Among these, the evaluation dimensions also vary by different degrees in various studies. It was established that the evaluation dimensions are largely based on economic, demographic, spatial, social, environmental, land, residents' lives, and sustainable development. In terms of empirical studies [21,22], there are also a significant number of studies globally, including studies on different research regions and research scales.

ESV, an important indicator of ecological environment quality, has received extensive academic attention. In 1997, Costanza et al. [23] proposed a global ES and natural capital value assessment method, which laid the foundation for the connotation between ESV and the assessment method. In addition, the work provided a methodological basis for subsequent studies. In 2003, Heights et al. [24–26] established a table of ecological service values per unit area for terrestrial ecosystems in China. The results showed that the equivalent factor method is widely used in Chinese academia. Evaluation methods include alternative market techniques [27] and simulated market techniques [28]. The alternative market technique expresses the economic value of ES in terms of "shadow prices" and consumer surplus. The simulated market technique (also known as the hypothetical market technique) measures ecosystem goods and services without market transactions and actual market prices by artificially constructing hypothetical markets. In today's academic community, in addition to varying evaluation methods, there are also some differences in terms of study areas and their scales. In terms of the scale of the study area, most scholars study at the municipal or provincial scale [29]. Other study subjects include the scale of economic circles [30], urban agglomerations [31], or water [32].

Globally, scholars have conducted extensive research regarding the interaction between NU and ESV and its CCD. Theoretical studies on the coupling relationship mainly include the Environmental Kuznets Curve hypothesis (EKC) [33,34], Telecoupling theory [35,36], Planetary boundaries theory [37,38], and Decoupling theory [39]. Regarding empirical studies, different domestic and foreign scholars have conducted a significant number of case studies regarding the relationship between NU and the global ecological environment [40–42]. These studies argue that there are significant interactions between NU and ESV. These different interactions lead to different development outcomes.

Based on considering the issues mentioned previously, this study selected Nanchang as the research area. Based on previous studies, first, the NU of Nanchang was evaluated in four dimensions: economic, demographic, spatial, and social. Second, ArcGIS was used to extract the area of each land use, and then ES was quantified using the equivalent factor table, and subsequently we conducted an analysis from the perspective of land use. Finally, the CD and CCD model was used to measure the differences between CD and CCD of NU and ES in space and time.

## 2. Materials and Methods

### 2.1. Study Area Overview and Data Sources

Study Area Overview

With a total population of about 6,255,000 and a total area of about 7195 square kilometers, Nanchang is the capital of Jiangxi Province. It is the political, economic, scientific, and cultural center of the province, and one of the 35 Chinese megacities. Nanchang is located in the central north of Jiangxi Province, in the alluvial plain of the Ganjiang River and the lower reaches of the Fuxiang River near Poyang Lake (Figure 1). It is located between 115°27′ and 116°35′ E and 28°10′ and 29°11′ N. The district has convenient and well-connected transportation, railroads, and highways. It is the only provincial capital city through which the Beijing–Kowloon Railway passes. The whole territory is mainly plain, being flat in the southeast and hilly in the northwest. Within the city, the average elevation is 25 m above sea level and contains a low-lying urban area with an average elevation of 22 m. In the west is the Xishan Mountain Range, with the highest point being the Meiling Main Peak Washing Medicine Dock at 841.4 m above sea level. The terrain is generally high in the northwest and low in the south and east, with low hills and plains developing in sequence, showing characteristics of layered landforms. With the Ganjiang River as the boundary, the northwestern part of the Ganjiang River is a tectonic denudation of low hills, and the east of the Ganjiang River is river erosion and accumulation plains, with rivers, lakes, harbors and branches, and braided water system development.

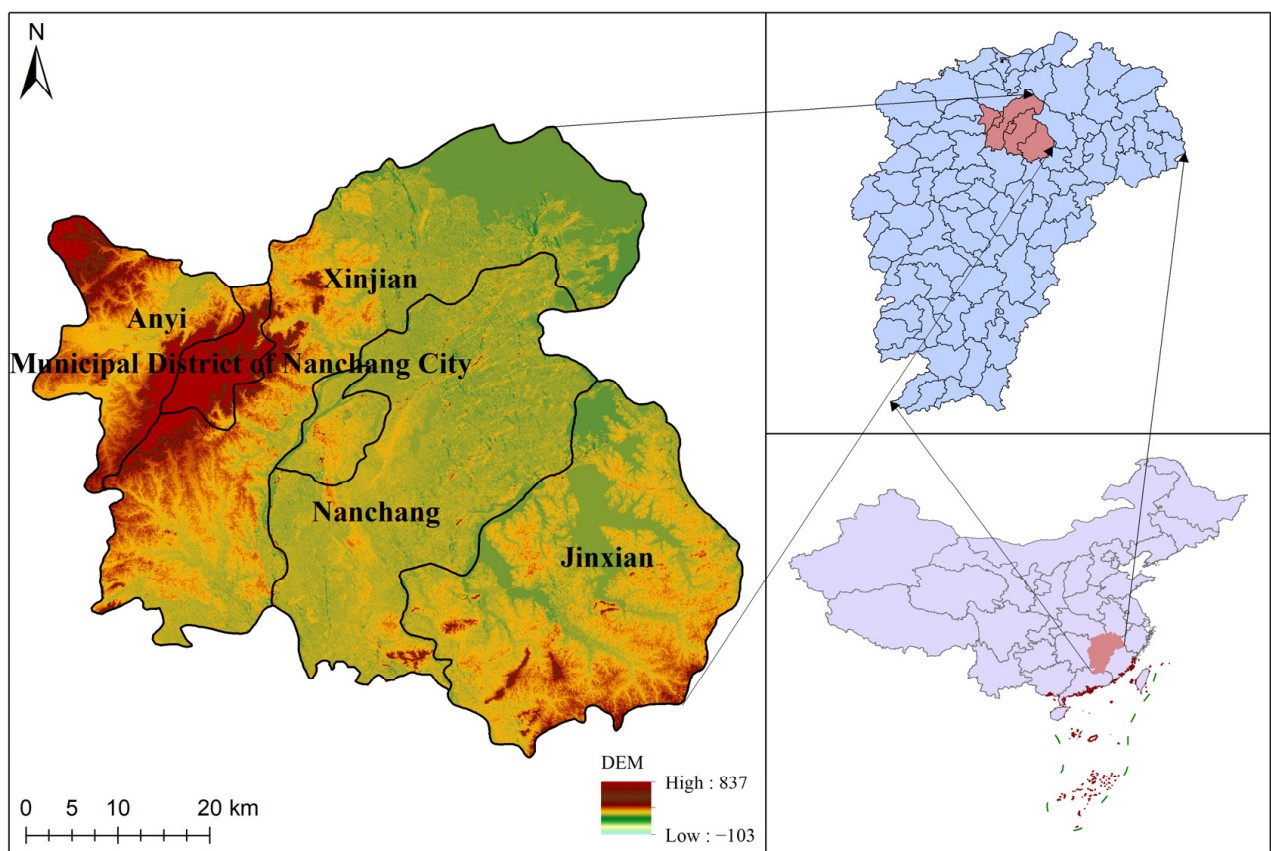

**Figure 1.** Map of the study area.

### 2.2. Data Sources

A research survey regarding NU and ESV in Nanchang was conducted. Data regarding land use and socioeconomic conditions in Nanchang for 2005, 2010, 2015, and 2020 were collected and compiled. Land use data production is produced by manual visual interpretation based on the previous year's data results using Landsat TM/ETM remote sensing

images of each period as the main data source, and the data format used in the study is ESRI GRID with a spatial resolution of 1 km and projection coordinates of Krasovsky 1940 Albers from the Data Center for Resource and Environmental Sciences, Chinese Academy of Sciences. The land use types included arable land, forest land, grassland, water, development land, and unused land. Socio-economic situation data were obtained from the Jiangxi Province Statistical Yearbook and the Nanchang National Economic and Social Development Bulletin.

### 2.3. NU Level Evaluation Methodology

Many methods exist to study NU, but most use the method of establishing indicator models. Indicator models can also be divided into single-indicator models and multi-dimensional comprehensive indicator models. Scholars have used different evaluation methods for different research purposes. In this study, based on previous studies, we adopted the evaluation method of constructing multi-dimensional comprehensive indicators. Considering the current situation in Nanchang and the availability of data, this study divides the NU into four subsystems: economic urbanization, population urbanization, spatial urbanization, and social urbanization. This may be further subdivided into 14 specific indicators, such as per capita gross regional product, the share of tertiary industry output in GDP, and social fixed asset investment. The four subsystems can be further subdivided into 14 specific indicators: per capita gross regional product, proportion of tertiary industry output in GDP, and social fixed asset investment. The specific indicators are listed in Table 1.

**Table 1.** New urbanization (NU) Indicator System.

| System Level | Guideline Level | Indicator Level | Indicator Type |
|---|---|---|---|
| **NU** | Economic urbanization | Per capita GDP (yuan/person) | Positive |
| | | The proportion of tertiary industry output value to GDP (%) | Positive |
| | | Total social fixed asset investment (RMB) | Positive |
| | | Value added of industrial enterprises above the scale | Positive |
| | Population urbanization | Proportion of urban population (%) | Positive |
| | | Proportion of population employed in tertiary industry | Positive |
| | | Population density (person/square kilometer) | Positive |
| | Spatial urbanization | Urban road area per capita (square meters) | Positive |
| | | Area of built-up area (square meters) | Positive |
| | | Urban population density (person/km$^2$) | Positive |
| | | Park green space per capita (square meters) | Positive |
| | Social urbanization | Number of public transportation operating vehicles (vehicles) | Positive |
| | | Number of hospital beds per 10,000 people (number) | Positive |
| | | Per capita disposable income of urban residents (yuan) | Positive |

Indicator Weight

While evaluating the comprehensive indexes, weights must be determined for each specific index. The methods of determining weights can be divided into subjective and objective weighting methods. Because the subjective ascertainment method relies on subjective evaluation and lacks objectivity, this study used the entropy value method for the objective weighting evaluation method. The specific steps of the entropy method [43] are as follows:

(1)    Standardization Standardization of positive indicators

$$x'_{ij} = \frac{x_{ij} - \min(x_{ij})}{\max(x_{ij}) - \min(x_{ij})} \tag{1}$$

Standardization of negative indicators

$$x'_{ij} = \frac{\max(x_{ij}) - x_{ij}}{\max(x_{ij}) - \min(x_{ij})} \tag{2}$$

where $x_{ij}$ represents the index value, $\max(x_{ij})$ represents the maximum value of the index, $\min(x_{ij})$ represents the minimum value of the index, $x'_{ij}$ represents the normalized index value, and $x'_{ij} \in [0, 1]$.

(2)  Determination of indicator weights: weight of jth item in ith region

$$p_{ij} = \frac{x'_{ij}}{\sum_{i=1}^{n} x'_{ij}} \tag{3}$$

(3)  Calculation of the entropy value of the jth indicator

$$e_j = -\frac{1}{\ln n} \sum_{i=1}^{n} p_{ij} \ln p_{ij} \tag{4}$$

(4)  Calculation of coefficient of variation

$$g_j = 1 - e_j \tag{5}$$

(5)  Calculate the weights of each indicator

$$w_j = \frac{g_j}{\sum_j g_j} \tag{6}$$

(6)  Calculate the composite score of the NU level for each city

$$F_i = \sum_j w_j x'_{ij} \tag{7}$$

*2.4. Land Use Dynamic Attitude*

The quantitative change in land-use types refers to the change in the total areas of different land-use types and the rate of change. The calculation formula is as follows:

$$L_k = \frac{L_b - L_a}{L_a} \times 100 \tag{8}$$

where $L_k$ denotes the total variation of a land type during the regional study period and $L_a$ and $L_b$ denote the area of a land type at the beginning and end of the study period, respectively.

*2.5. ESV Evaluation Method*

In this study, the ESV was evaluated using the equivalence factor method, where one standard unit of the ecosystem ecological service value equivalence factor is the economic value of the annual natural food production of 1 hm$^2$ of farmland with a national average yield [44]. The economic value of one ecological service value equivalent factor is equal to 1/7 of the market value of the national average grain yield in that year [25]. Therefore, in this study, we calculated the land ESV equivalent based on 1/7 of the unit area output value of the total food production value in Nanchang during 2020 and obtained the ecological service value of each equivalent of land ecosystem in Nanchang as 2194.99 yuan/hm$^2$ to

obtain the ESV table of the unit area in the study area, as shown in Table 2. The ESV of each prefecture-level city in the Jiangxi Province was calculated using the following equation:

$$ESV = \sum A_k \times VC_k \tag{9}$$

where ESV is the ecosystem service value, yuan, $A_k$ is the area of the kth land-use type in the study area, $hm^2$, and $VC_k$ is the ESV coefficient, which is the ESV per unit area of the kth land-use type.

**Table 2.** Unit area of ecosystem service value (ESV).

| First Classification | Second Classification | Arable Land | Forest Land | Grassland | Water | Unused Land | Development Land | Total |
|---|---|---|---|---|---|---|---|---|
| Supply Services | Food production | 4851 | 2217 | 1536 | 1756 | 22 | 0 | 10,382 |
| | Raw material production | 1076 | 5092 | 2261 | 505 | 66 | 0 | 8999 |
| | Water supply | −5729 | 2634 | 1251 | 22,938 | 44 | 0 | 21,138 |
| Regulation Services | Gas regulation | 3907 | 16,748 | 7946 | 2085 | 285 | 0 | 30,971 |
| | Climate regulation | 2041 | 50,112 | 21,006 | 6212 | 219 | 0 | 79,590 |
| | Purification of the environment | 593 | 14,684 | 6936 | 12,533 | 900 | 0 | 35,647 |
| | Hydrological regulation | 6563 | 32,793 | 15,387 | 240,066 | 527 | 0 | 295,336 |
| Support Services | Soil conservation | 2283 | 20,391 | 9680 | 2041 | 329 | 0 | 34,725 |
| | Maintenance of nutrient cycles | 680 | 1558 | 746 | 154 | 22 | 0 | 3161 |
| | Biodiversity | 746 | 18,570 | 8802 | 5619 | 307 | 0 | 34,044 |
| Cultural Services | Aesthetic landscape | 329 | 8143 | 3885 | 4346 | 132 | 0 | 16,836 |
| | **Total** | 17,340 | 172,943 | 79,437 | 298,256 | 2853 | 0 | 570,830 |

### 2.6. CCD Evaluation Method

Coupling is a concept in physics, and CD refers to a phenomenon in which two or more systems or two forms of motion interact or influence each other. In contrast, CCD is capable of quantifying the strength of coordination between systems in addition to interaction. Consequently, this paper uses CCD to assess the degree of coordination between the new township and ES. It can also be used to judge the CD criteria which are shown in Tables 3 and 4. The formulae for the CD and CCD are as follows:

$$C = 2\sqrt{\frac{u_1 \times u_2}{(u_1 + u_2)^2}} \tag{10}$$

where C is the coupling degree, $u_1$ is the standardized NU level, and $u_2$ is the standardized ESV,

$$D = \sqrt{C \times T} \tag{11}$$

$$T = \alpha u_1 + \beta u_2 \tag{12}$$

where D is the coupling coordination degree, T is the comprehensive coordination index, and $\alpha$ and $\beta$ are the contribution coefficients, both of which were taken as 0.5.

**Table 3.** Coupling degree (CD) judgment criteria.

| CD | Stage |
|---|---|
| C = 0 | Disorderly development stage |
| C ≤ 0.3 | Low-level coupling stage |
| 0.3 < C ≤ 0.5 | Stubborn stage |
| 0.5 < C ≤ 0.8 | Breaking-in stage |
| 0.8 < C ≤ 1 | High-level coupling stage |

**Table 4.** CCD judging standards.

| CCD | Stage | CCD | Stage |
|---|---|---|---|
| 0–0.09 | Extreme disorder | 0.50–0.59 | Barely coordinated |
| 0.10–0.19 | Severe disorders | 0.60–0.69 | primary coordination |
| 0.20–0.29 | Moderate disorder | 0.70–0.79 | Intermediate coordination |
| 0.30–0.39 | Mild disorder | 0.80–0.89 | Good coordination |
| 0.40–0.49 | Imminent disorder | 0.90–1.00 | Excellent coordination |

## 3. Results

### 3.1. Characteristics of Evolving NU Level

NU refers to the development of the urban economy, urban population increase, expansion of urban land and space, and improvement of the urban living environment in terms of comfort and convenience. Therefore, according to the definition of NU, this study measured the NU development level in four dimensions: economic, demographic, spatial, and social.

As shown in Table 5, the NU level in Nanchang city increased from 0.0176 in 2005 to 0.9493 in 2020 with a yearly increasing trend. This indicates that the Nanchang city NU is developing faster, and the scale of the city is continuously expanding. The Nanchang development has entered the middle and late regime of rapid NU development, showing the characteristics of changing from scale expansion to both scale and quality, from single-unit development to urban clusters, from single economic goal to comprehensive development, and from single mode to specialization. Consequently, the year-by-year increasing NU level indicates that Nanchang has performed a more successful transformation of NU development.

**Table 5.** Nanchang city NU level.

| Subsystems | Year | 2005 | 2010 | 2015 | 2020 | Weights |
|---|---|---|---|---|---|---|
| Economic urbanization | Per capita GDP (yuan/person) | 22,390 | 43,961 | 75,879 | 92,697 | 0.0657 |
| | The proportion of tertiary industry output value to GDP (%) | 40 | 37.8 | 41.2 | 49.3 | 0.0915 |
| | Total social fixed asset investment (million yuan) | 484,0586 | 19,233,503 | 40,000,719 | 68,014,781 | 0.0746 |
| | Value added of industrial enterprises above the scale | 2,311,874 | 6,509,234 | 14,518,438 | 21,950,086 | 0.0750 |
| Population urbanization | Proportion of urban population (%) | 44.88 | 65.71 | 71.56 | 78.08 | 0.0523 |
| | Proportion of population employed in tertiary industry | 39.1 | 50.9 | 43.8 | 49.7 | 0.0605 |
| | Population density (person/km$^2$) | 603 | 646 | 737 | 869 | 0.0842 |
| Spatial urbanization | Urban road area per capita (m$^2$) | 7.78 | 8.5 | 12.75 | 11.34 | 0.0833 |
| | Urban population density (person/km$^2$) | 3476 | 9881 | 7536 | 7363 | 0.0540 |
| | Area of built-up area (m$^2$) | 134.97 | 201.5 | 307.3 | 366.02 | 0.0672 |
| | Park green space per capita (m$^2$) | 7.3 | 9.01 | 11.8 | 12.27 | 0.0638 |
| Social urbanization | Number of public transportation operating vehicles (vehicles) | 1824 | 2490 | 3377 | 4381 | 0.0705 |
| | Number of hospital beds per 10,000 people (number) | 15,066 | 20,025 | 30,169 | 44,206 | 0.0826 |
| | Per capita disposable income of urban residents (yuan) | 10,301.28 | 18,276 | 31,942 | 46,796 | 0.0749 |
| **NU** | | 0.0176 | 0.3163 | 0.6293 | 0.9493 | |
| **Economic urbanization** | | 0.017535 | 0.053230 | 0.165246 | 0.306887 | 0.3069 |
| **Population urbanization** | | 0.000020 | 0.106896 | 0.108591 | 0.190986 | 0.1970 |
| **Spatial urbanization** | | 0.000027 | 0.107349 | 0.225364 | 0.223396 | 0.2682 |
| **Social urbanization** | | 0.000023 | 0.048805 | 0.130054 | 0.227997 | 0.2280 |

As shown in Figure 2, the temporal development characteristics of each NU subsystem shows an annual increase, indicating that each NU subsystem in Nanchang city is progressing and developing in tandem. A comparative study of the subsystems indicates that the rate and level of development of each subsystem are not consistent. As shown in Table 5, the level of economic urbanization ranked first at the beginning of the study, while the other three subsystems were comparable. During the study, the level of economic urbanization still ranked first, but the difference between the other subsystems gradually increased. Spatial and social urbanization closely followed. At the beginning of the study, spatial urbanization was ahead of social urbanization, but in 2020, social urbanization surpassed spatial urbanization. This indicated that Nanchang city notably progressed during urbanization. This progress not only considered spatial expansion, but also included social amenities. In terms of the weight of each subsystem, economic urbanization and spatial urbanization are the highest ranked. The development level of each subsystem and their respective weights were greater, and the actual development level of economic urbanization and spatial urbanization were also greater. To realize balanced NU level development, the development focus can be considered to shift to the other two subsystems in future development.

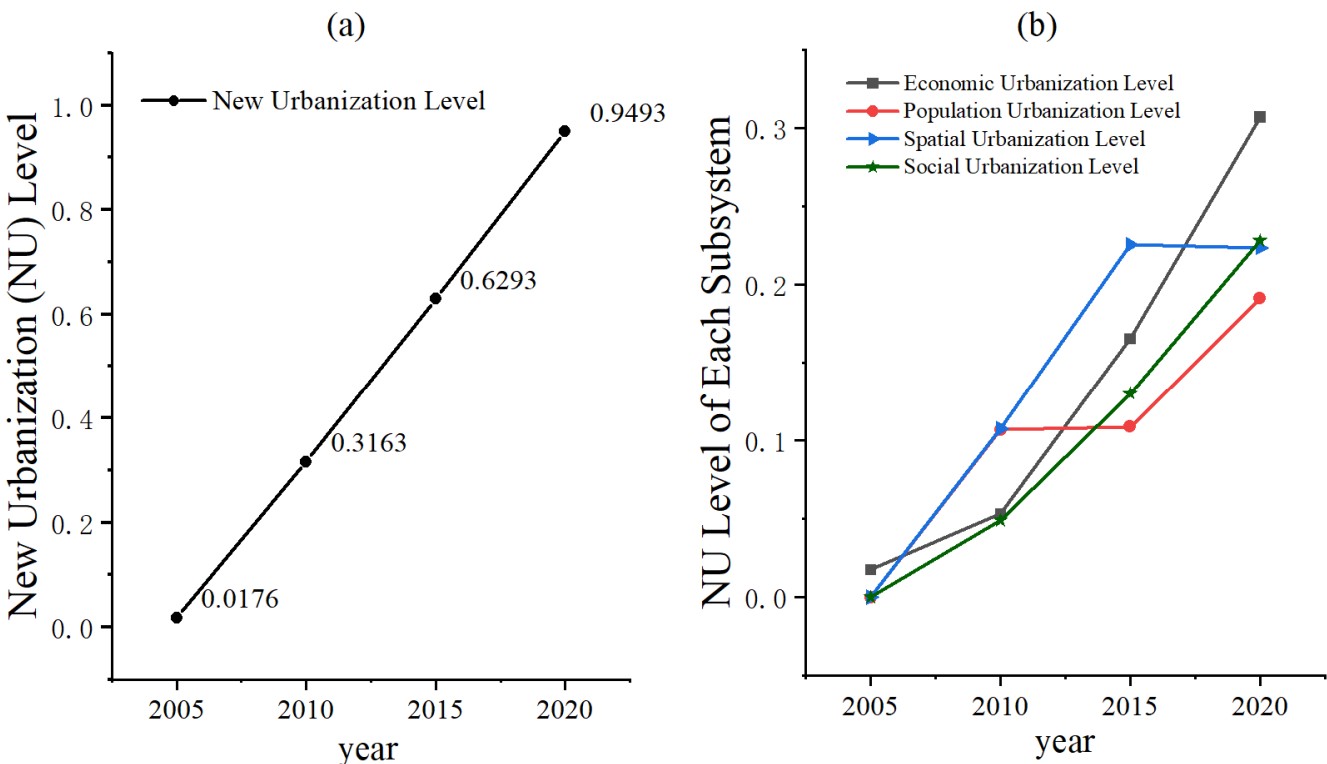

**Figure 2.** (**a**) Change in NU level; (**b**) change in NU level per each subsystem.

### 3.2. Characteristics of Land Use Dynamic Attitude Change

Statistical analysis of land use in Nanchang city was performed, and the results are shown in Table 6. From 2005 to 2020 the development land experienced the greatest magnitude of change, which was continuous, with a total increase of 21,300 hm². The increase was followed by arable land, whose area continues to decrease, with a total decrease of 10,900 hm². The forest land area increased by 0.01 million hm² from 2005 to 2010, and then continued to decrease, and overall, the area decreased by a total of 0.39 million hm² during the study period. Both forest land and unused land increased and then decreased. The unused land area decreased by a total of 0.07 million hm² from 2005 levels. From 2005 to 2010, the area of unused land increased by 0.42 × 10⁴ hm². From 2010 to 2020, the area of unused land continued to decrease, by 0.49 × 10⁴ hm². The grassland area continued to decrease, but the overall change was not significant, with a total

decrease of only $0.23 \times 10^4$ hm$^2$. The change in water area showed fluctuating increases and decreases, with a net decrease of $0.35 \times 10^4$ hm$^2$. The water area had a net decrease by $0.35 \times 10^4$ hm$^2$. From the structure of land types in Nanchang, arable land is the most widely distributed, followed by forest land and water, accounting for greater than 80% of the overall area of Nanchang. In terms of dynamic attitude change, the greatest change is development land in 2015–2020 with a dynamic change of 3.59%, followed by development land in 2010–2015 with a dynamic change of 3.44%. The smallest change is for forest land in 2005–2010 with a dynamic behavior. The minimal change is for forest land from 2005 to 2010, with a dynamic change of 0.016%.

**Table 6.** Area and dynamic attitude of Nanchang city by year.

| Year | Type | Arable Land | Forest Land | Grassland | Water | Development Land | Unused Land |
|---|---|---|---|---|---|---|---|
| 2005 | Area ($10^4$ hm$^2$) | 38.99 | 12.24 | 0.90 | 11.68 | 5.03 | 3.20 |
|  | Proportion (%) | 54.93 | 16.79 | 1.23 | 15.81 | 6.83 | 4.40 |
| 2010 | Area ($10^4$ hm$^2$) | 38.95 | 12.25 | 0.82 | 11.22 | 5.18 | 3.62 |
|  | Proportion (%) | 54.92 | 16.83 | 1.13 | 15.09 | 7.02 | 5.01 |
| 2015 | Area ($10^4$ hm$^2$) | 38.35 | 12.03 | 0.80 | 11.31 | 6.07 | 3.48 |
|  | Proportion (%) | 54.09 | 16.51 | 1.10 | 15.23 | 8.25 | 4.83 |
| 2020 | Area ($10^4$ hm$^2$) | 37.90 | 11.85 | 0.67 | 11.33 | 7.16 | 3.13 |
|  | Proportion (%) | 52.61 | 16.45 | 0.93 | 15.73 | 9.94 | 4.34 |
| 2005–2010 | Area of change ($10^4$ hm$^2$) | −0.04 | 0.01 | −0.08 | −0.46 | 0.15 | 0.42 |
|  | Dynamic attitude (%) | 0.021 | 0.02 | −1.78 | −0.79 | 0.60 | 2.63 |
| 2010–2015 | Area of change ($10^4$ hm$^2$) | −0.60 | −0.22 | −0.02 | 0.09 | 0.89 | −0.14 |
|  | Dynamic attitude (%) | −0.31 | −0.36 | −0.49 | 0.16 | 3.44 | −0.77 |
| 2015–2020 | Area of change ($10^4$ hm$^2$) | −0.45 | −0.18 | −0.13 | 0.02 | 1.09 | −0.35 |
|  | Dynamic attitude (%) | −0.23 | −0.30 | −3.25 | 0.04 | 3.59 | −2.01 |

*3.3. Land Use Changes*

Land use changes are shown in Figure 3, where land shifts at the beginning of the study were not significant and from 2015 to 2020, land use changes changed significantly. The specific changes for each land type are as follows.

The area of arable land has been decreasing (Figure 3). Between 2005 and 2010, a slight area decrease occurred, with a transfer of 3500 hm$^2$ from arable land to other types and 3100 hm$^2$ from other types to arable land. Between 2010 and 2015, 6200 hm$^2$ were transferred from arable land to development land, and not to other types, while only 200 hm$^2$ were transferred from forest land to arable land. Therefore, overall, forest land experienced some large declines during this period. During 2015–2020, arable land was transferred out to other land types by 102,800 hm$^2$, mainly to forest land, water and development land. Other types contributed 98,300 hm$^2$ into arable land, mainly from forest land and water. Overall, the area decreased by 4500 hm$^2$.

The area of forest land first increased and then decreased (Figure 3). Between 2005 and 2010, 1700 hm$^2$ were transferred to other types, but 1800 hm$^2$ were also transferred to forest land from other types; consequently, a slight increase in forest land occurred during this period. Between 2010 and 2015, no other types of land were transferred; however, 2300 hm$^2$ of forest land was transferred outward, mostly to development land. During 2015–2020, there was a significant change in land use, with 50,300 hm$^2$ being transferred from forest land, most of which was transfer to arable land. An area of 48,500 hm$^2$ were transferred from other types, mostly from arable land to forest land. Although the transfer in and out of forest land has changed significantly, the overall area of forest land decreased by 1800 hm$^2$, which was not a significant change.

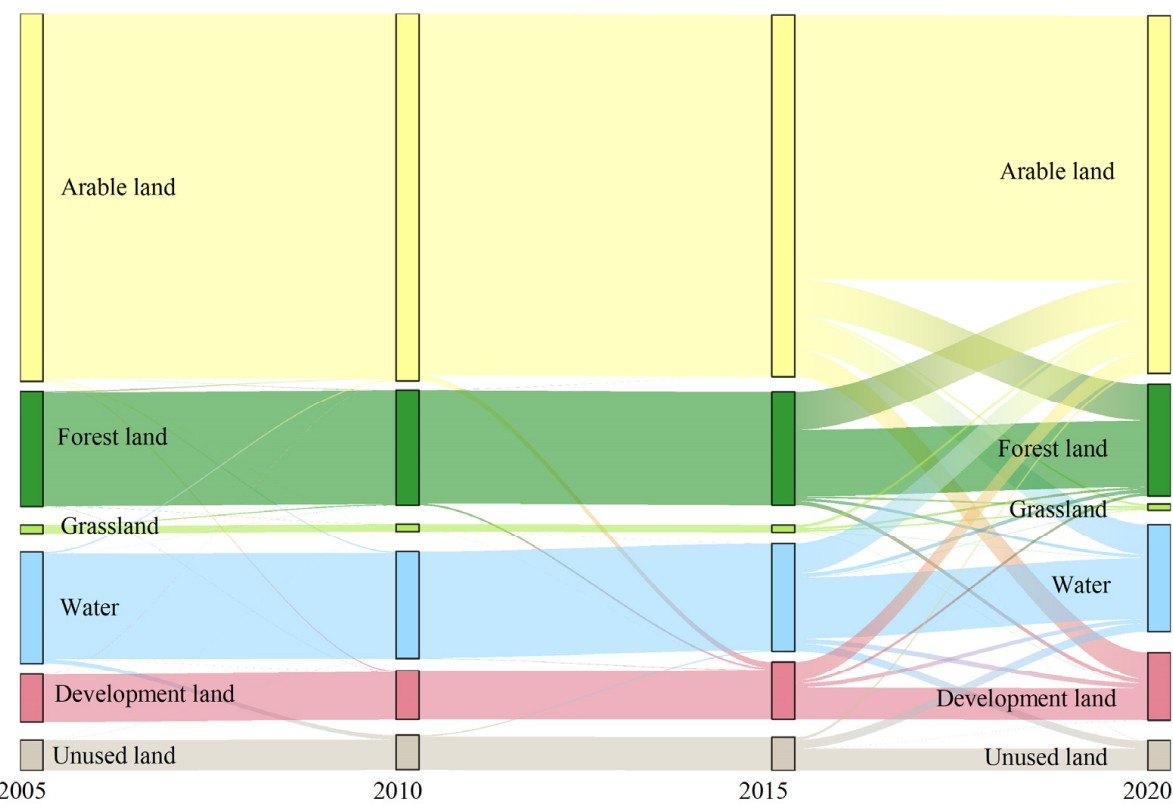

**Figure 3.** Land use area change map.

Grassland area decreased during the study period (Figure 3). Between 2005 and 2010, grassland transferred 1200 hm² of land to the forest category. Other types transferred 400 hm² to grassland, resulting in a net decrease of 800 hm². Between 2010 and 2015, grasslands transferred 200 hm² to other types. Other types did not transfer to grasslands, resulting in a total decrease of 200 hm². Between 2015 and 2020, grassland transferred 6200 hm² to other land types. The other types transferred 4900 hm² to grasslands, resulting in a net decrease of 1300 hm².

The water area initially had a fluctuating decrease, followed by an increase, and finally decreased during the study period (Figure 3). During 2005 and 2010, 6300 hm² were transferred out of the water and 1, 700 hm² were transferred in. During 2010 and 2015, 600 hm² were transferred out of the water and 1500 hm² were transferred in. Between 2015 and 2020, 49,200 hm² were transferred out of the water and 49,400 hm² were transferred in. During the study period, there was a net decrease of 3500 hm².

The area of land development has continued to increase (Figure 3). Between 2005 and 2010, development land transferred 200 hm² to arable land. Whereas other types of land transferred 1700 hm² to development land. Between 2010 and 2015, development land did not transfer to other types of land while other land types transferred 8900 hm² in. Between 2015 and 2020, 27,400 hm² of development land was transferred to other land types and 38,300 hm² of other types of land were transferred to development land. The total area increased from 50,300 hm² to 71,600 hm².

The unused land area initially increased and subsequently decreased (Figure 3). Between 2005 and 2010, 300 hm² of unused water were transferred outward and 4500 hm² of water were transferred to them. Between 2010 and 2015, there is no other type of land transfer except for the transfer of 1400 hm² of unused land to waters. Between 2015 and 2020, unused land transferred 11,900 hm² to other types of land, and other land types transferred 8400 hm² to unused lands. The area of unused land had a net decrease of 700 hm². This was a relatively insignificant change.

The principal causes for these changes include: (1) a series of policies and behaviors, including the implementation of the project of returning farmland to forest in Nanchang, the expansion of urban areas to expropriate arable land in the suburbs, and illegal land abuse in villages and towns (e.g., illegal occupation of arable land to build houses, soil extraction, and mining of arable land). These policies led to a continuous decline of arable land during the study period. (2) The area of forest land initially increased and subsequently decreased. In general, the area decreased. At the beginning of the study, the green belt had a net transformation from grassland to forest land because trees were considered greener and more beautifying than grass. In addition, the cost and maintenance of grassland was greater. Later, the area of forest land decreased due to deforestation and conversion to arable land or other sites. (3) The continuous reduction in grassland area due to the transformation of the green belt and the reclamation of grassland into arable land. (4) During the early stage of the study, there was urban expansion in Nanchang city, such as filling the lake and creating land around the lake for forestation; consequently, the water area was reduced. To protect the ecological environment and prevent further decreases in water area, Jiangxi Province has adopted policies and regulations such as the "Regulations on Environmental Protection of Poyang Lake Ecological and Economic Zone," "Notice of the General Office of Nanchang Municipal People's Government on Strictly Prohibiting the Enclosure of Lakes in Poyang Lake Area", and "Regulations on Lake Protection in Jiangxi Province (Draft)." These policies prohibit many activities which divide and encroach upon water surface, so the area of water has continued to increase subsequent to 2010. (5) Due to the accelerated NU process, urban land is continuously expanding, resulting in a continuous increase in land development.

### 3.4. Spatial and Temporal ESV Evolutionary Characteristics
### 3.4.1. Characteristics of Quantitative ESV Changes

As shown in Figure 4, the Nanchang city ESV continued to decline during the study period, from an initial 63.572 hundred million yuan to a final 61.480 hundred million yuan at the end of the study period (Table 7), a decline of 2.092 hundred million yuan over 15 years. Among the ESV decline, water and forest land were the main ESV components, with a total share greater than 87%. Water was the greatest contributor to all land types, accounting for approximately 53%. The water area decreased by 3500 hm$^2$; however, its ESV contribution was the greatest (with a significant difference over the second greatest contributor). The water area contribution was a result of its elevated unit area value and its third-place ranking for total area among all land types.

Panels a,b of Figure 4 show that ES types can be divided into supply, regulation, support, and cultural services. Numerically, the ESV of regulation services was the largest among all service types. Its value during the study period varied from 636 hundred million yuan to 615 hundred million yuan. The ESV of regulation services is the greatest among all service types, with a value greater than 54.7 hundred million yuan and a 78% share of net ESV. The ESVs of the other three service types, in descending order, were support services, supply services, and cultural services. In terms of time series, all service types showed a gradual decrease over the study period.

In terms of the ESV of individual services (Table 8), the descending order of each service type was regulation services, support services, supply services, and cultural services. Among these, the single services that contribute the most are hydrological regulation, climate regulation, gas regulation, soil conservation, environmental purification, and biodiversity. Maintenance of the nutrient cycle was the minimal contribution among all individual services, accounting for only 0.76%. The value of each individual service showed a continuously decreasing trend during the study period, except for water supply and hydrological regulation. For these water supply services first decreased and then increased. Hydrological regulation showed highly variable changes.

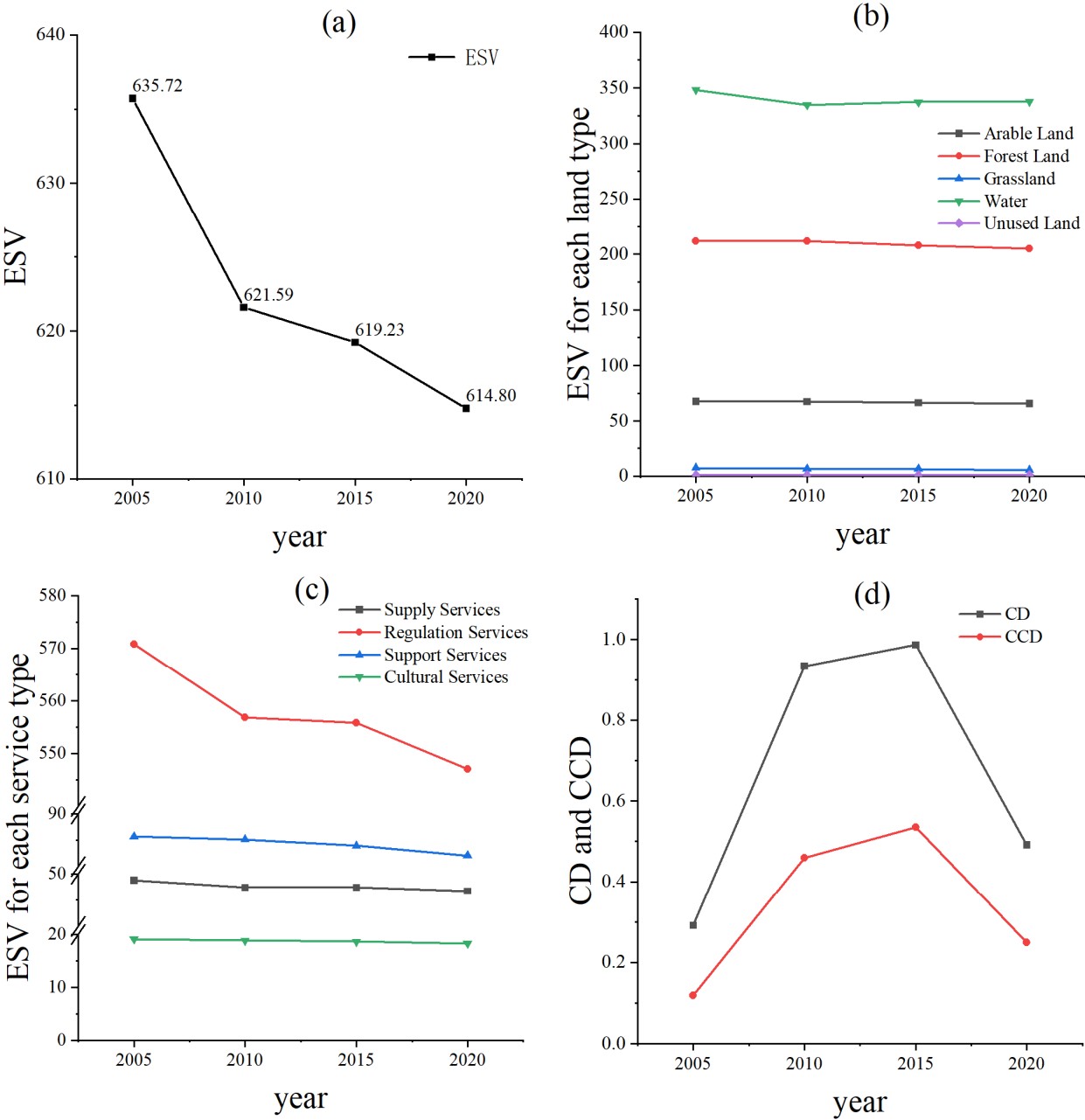

**Figure 4.** (**a**) Change in ESV, (**b**) change in ESV per land type, (**c**) change in ESV per service type, (**d**) change in coupling degree (CD) and coupling coordination degree (CCD).

**Table 7.** ESV for each land use type.

| Land Use Type | 2005 | | 2010 | | 2015 | | 2020 | |
|---|---|---|---|---|---|---|---|---|
| | Value (Hundred Million Yuan) | Percentage (%) | Value (Hundred Million Yuan) | Percentage (%) | Value (Hundred Million Yuan) | Percentage (%) | Value (Hundred Million Yuan) | Percentage (%) |
| Arable land | 67.61 | 10.64 | 67.54 | 10.87 | 66.50 | 10.74 | 65.72 | 10.69 |
| Forest land | 211.68 | 33.30 | 211.86 | 34.08 | 208.05 | 33.60 | 204.94 | 33.33 |
| Grassland | 7.15 | 1.12 | 6.51 | 1.05 | 6.35 | 1.03 | 5.32 | 0.87 |
| Water | 348.36 | 54.80 | 334.64 | 53.84 | 337.33 | 54.48 | 337.92 | 54.97 |
| Unused land | 0.91 | 0.14 | 1.03 | 0.17 | 0.99 | 0.16 | 0.89 | 0.15 |
| Development Land | 0.00 | 0.00 | 0.00 | 0.00 | 0.00 | 0.00 | 0.00 | 0.00 |
| Total | 635.72 | 100 | 621.59 | 100 | 619.23 | 100 | 614.80 | 100 |

**Table 8.** ESV for each service type.

| ES | | 2005 | | 2010 | | 2015 | | 2020 | |
|---|---|---|---|---|---|---|---|---|---|
| | | Value (Hundred Million Yuan) | Percentage (%) | Value (Hundred Million Yuan) | Percentage (%) | Value (Hundred Million Yuan) | Percentage (%) | Value (Hundred Million Yuan) | Percentage (%) |
| Supply Services | Food production | 23.82 | 3.75 | 23.71 | 3.82 | 23.39 | 3.78 | 23.11 | 3.76 |
| | Raw material production | 11.24 | 1.77 | 11.20 | 1.80 | 11.03 | 1.78 | 10.85 | 1.77 |
| | Water supply | 7.80 | 1.23 | 6.77 | 1.09 | 7.26 | 1.17 | 7.49 | 1.22 |
| Regulation Services | Gas regulation | 38.98 | 6.13 | 38.83 | 6.25 | 38.22 | 6.17 | 37.64 | 6.12 |
| | Climate regulation | 78.51 | 12.35 | 78.11 | 12.57 | 76.90 | 12.42 | 75.63 | 12.30 |
| | Purification of the environment | 35.84 | 5.64 | 35.25 | 5.67 | 34.98 | 5.65 | 34.59 | 5.63 |
| | Hydrological regulation | 347.68 | 54.69 | 336.54 | 54.14 | 337.55 | 54.51 | 336.92 | 54.80 |
| Support Services | Soil conservation | 37.22 | 5.85 | 37.07 | 5.96 | 36.48 | 5.89 | 35.88 | 5.84 |
| | Maintenance of nutrient cycles | 4.81 | 0.76 | 4.80 | 0.77 | 4.73 | 0.76 | 4.66 | 0.76 |
| | Biodiversity | 33.09 | 5.21 | 32.79 | 5.28 | 32.37 | 5.23 | 31.89 | 5.19 |
| Cultural Services | Aesthetic landscape | 16.72 | 2.63 | 16.50 | 2.65 | 16.33 | 2.64 | 16.12 | 2.62 |
| **Total** | | 635.72 | | 621.59 | | 619.23 | | 614.80 | |

### 3.4.2. Spatial and Temporal Variation Characteristics of ES Functional Values Based on Grid Scale

The spatial distribution of ESV within Nanchang city was visualized using a 1 km × 1 km grid unit. As shown in Figure 5 the elevated ESV area in Nanchang is principally located in the northeast corner near Poyang Lake and in the southeast area near the Junshan, Qinglan, and Jinxi lakes. The depleted ESV area in Nanchang is principally located in the central part of Nanchang county, where the main land use type is arable.

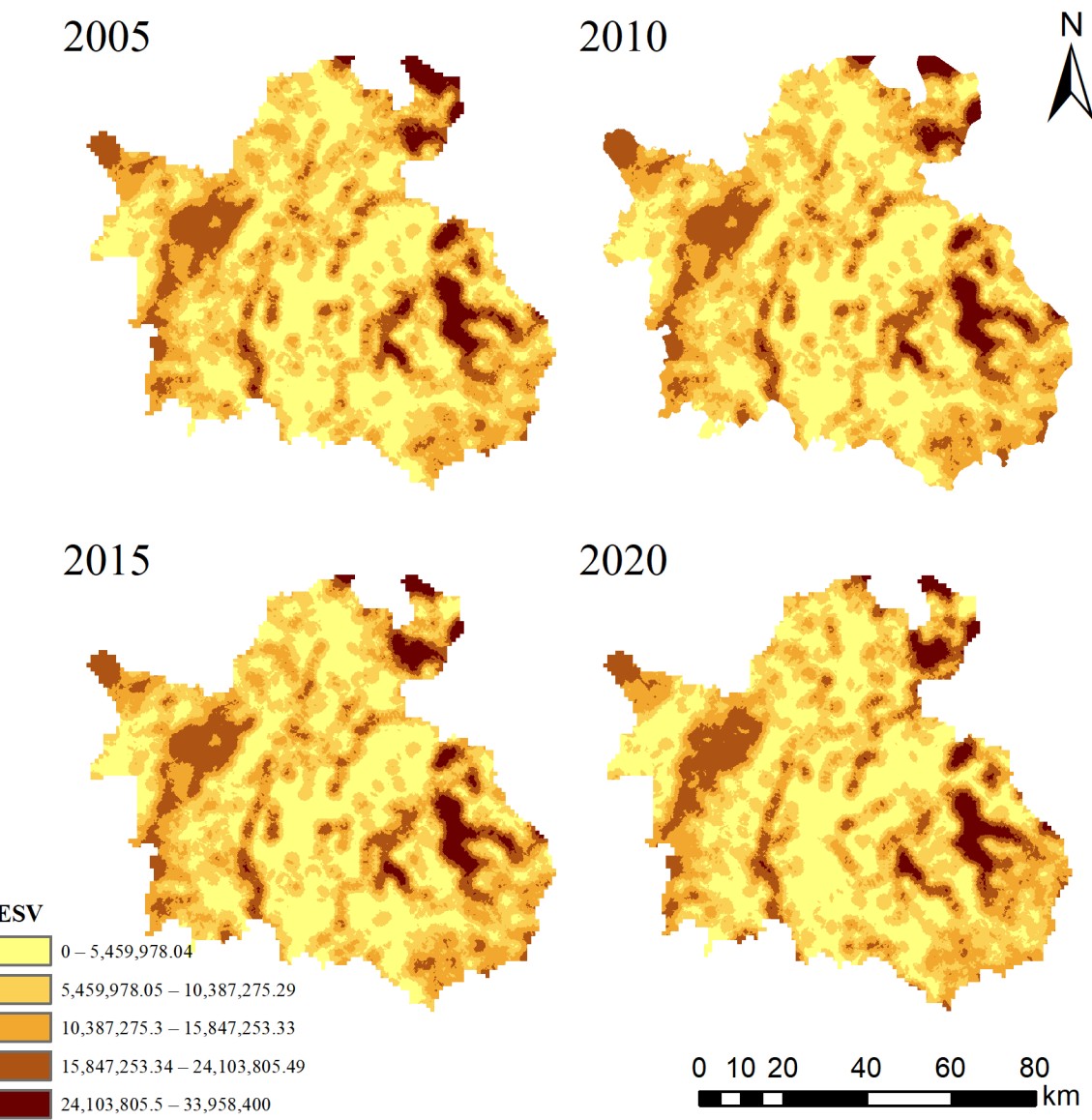

**Figure 5.** Spatial distribution of the ESV grid scale within Nanchang city.

The distribution of the grid-scale ecological high-value area in Nanchang city showed a net decrease during the study period. The high-value area in the northeast corner of Nanchang city is segmented. This is mainly because the Poyang Lake area in this area is decreasing. The land use type is changing from water to arable land, which results in an ESV decrease within this grid. In addition, the distribution of high ecological value areas in the northwest of Nanchang city is expanding. This expanding area includes the Meiling National Forest Park, because part of the grassland in this area is transformed into forest land. This results in an outward expansion of the distribution of areas with high ecological value.

*3.5. Characteristics of Coupled Changes Relating to NU Level and ES Capacity*

From Figure 5, the CD of the NU level and ES capacity in Nanchang city shows an inverted U-shaped development trend, initially increasing and subsequently decreasing. Between 2005–2015, the CD increases from low-level coupling stage to high-level coupling stage, and then falls to stubborn stage in 2020 (Table 9). This is mainly because the ecological evaluation index of Nanchang city was greater than the NU evaluation index at the initial stage, while the evaluation of the NU index continued to increase. The ecological evaluation index continued to decrease during the study period. The relative gap between the two shrinks increased. This resulted in the CD initially increasing and subsequently decreasing.

**Table 9.** CD and CCD by year and stage.

| Year | CD | Stage | CCD | Stage |
|------|------|-------|------|-------|
| 2005 | 0.291457 | Low-level coupling stage | 0.118180 | Severely disordered |
| 2010 | 0.934240 | High-level coupling stage | 0.459282 | Imminent disorder |
| 2015 | 0.986918 | High-level coupling stage | 0.534802 | Barely coordinated |
| 2020 | 0.491240 | Stubborn stage | 0.249231 | Moderately disordered |

The CCD for NU and ES capacity within Nanchang also showed an inverted U-shape, initially increasing and subsequently decreasing. There were fluctuations within the evolutionary process. From 2005 to 2015, CCD gradually increased and transitioned from severely disordered to barely coordinated. However, between 2015 and 2020, Nanchang city experienced a CCD regression devolving from the barely coordinated stage in 2015 to the moderately disordered stage. This indicated that the rapid NU advancement produced greater pressure on the ecological environment, causing the CCD to decline. In the future, Nanchang city should consider appropriately slowing down the NU process and promoting urban development without damaging the ecological environment.

## 4. Discussion

To provide a new understanding of the coupling coordination relationship between NU and ESV, the spatiotemporal distribution changes in NU, ESV and their CCD in Nanchang were investigated. From the results of the study, it is shown that NU gradually increases, ESV gradually decreases, and CCD ascends and then decreases, during the study period.

Specifically, the overall NU of Nanchang is gradually rising, and all subsystems are also on an upward trend, but the growth pace of each subsystem varies in different periods. The spatial and population subsystem of NU is higher at the beginning of the study and slowly lags behind afterwards, indicating that the development of urbanization has a certain lag. The initial phase of urbanization development is driven by the development of spatial urbanization and population urbanization first, that is, the development of urban land and the influx of new population in the city to drive the development of the other two subsystems, while at the later stage of the study, the economy has been developed and the respective conveniences of the city are equipped in turn due to the previous construction, indicating that the level of urbanization in Nanchang is now out of the initial development phase. Since urbanization is now a new type of urbanization, requiring high quality development, the previous rough development model is no longer applicable, so relying on the expansion of urban construction land to improve spatial urbanization is obviously not in line with today's development goals, and the unstable income due to the epidemic, coupled with the implementation of the rural revitalization strategy, has slowed down the pace of farmers moving to the city, resulting in the development of population urbanization is not like before. Therefore, to enhance the level of new urbanization, it should be considered from two aspects: economic urbanization and social urbanization.

The general ESV of Nanchang shows a gradual decreasing trend, but there are both increasing and decreasing ESVs in different regions under the grid scale. This is mainly due to the rapid urbanization that has led to changes in the status of land use in Nanchang,

the most significant of which is the increase in development land and the decrease in arable land, and other land types have also decreased in area. The overall level of ESV shows a decreasing trend, but from the grid scale, there is an increase and a decrease. The most significant changes are Meiling National Forest Park in the northwest and part of Poyang Lake location in the northeast corner, Meiling National Forest Park shows an increase in ESV because it is well protected, while part of Poyang Lake area now suffers from soil erosion, vegetation destruction, wetland degradation, and other problems that make the ESV in the local area decrease. This indicates that Nanchang's land use is unevenly distributed and can be zoned, and then appropriate development policies can be adopted for different zones to enhance ESV.

The CCD showed a trend of increasing and then decreasing, mainly because the ESV was higher than the NU in the early period, while the NU gradually increased and the ESV gradually decreased to less than the NU in the later period, so the difference between the two decreased and then increased, resulting in the CCD increasing and then decreasing. The results of Yanpeng Gao (2021) [45], Jianji Zhao (2020) [46], and others are roughly the same as this paper, and the coupling coordination eventually decreases. Han Han (2019) [32], Kaize Zhang (2019) [47], Weijie Li (2021) [42], Xiaoxue Ma (2022) [11], Chao Liu (2021) [48], and others showed an upward trend at the later stage, unlike this paper. Summarizing the above studies, it can be found that the study areas with the same coupling coordination degree eventually decreasing as this study are Liaoning Province and Yellow River Basin, respectively, and Nanchang City, like them, is a relatively economically underdeveloped area in China, while the study areas with the coupling coordination degree eventually increasing are Chongqing, Nanjing, Beijing, Yangtze River Basin Economic Zone, and China's coastal zones, which are economically highly developed regions. The highly developed urban areas have long passed the stage of sacrificing ecology for economic development, and are able to integrate various resources well while achieving urbanization development and protecting ecological environment. On the other hand, the economically less developed areas, due to their relatively backward development and their respective geographical environment, have a sloppy development and a more homogeneous industrial structure, which have an increasingly serious pressure on the ecological environment, thus leading to an eventual decline in coupling coordination. In addition, some studies have shown that a high level of NU has a positive effect on the ecological environment [18,49], so in order to make the CCD improve, we should focus on improving ESV while ensuring a high level of quality NU.

## 5. Conclusions

### 5.1. Conclusions

This study uses Nanchang city as the research object, based on land use data and socio-economic data from 2005, 2010, 2015, and 2020. In addition, it uses Geographical Information Systems (GIS) technology to indicate the changes in land use in Nanchang city by using land use quantity change and the land use transfer matrix. The study evaluates ESV from the overall scale and grid scale of Nanchang city by the equivalent factor method. Subsequently the CCD model was used to evaluate the CCD between NU and ES. The results are as follows.

(1) The NU within Nanchang shows an annual increase; the NU is developing faster while the spatial scale of the city is expanding continuously. Among the subsystems of NU, economy and spatial urbanization were the primary subsystems that were greater than the remaining subsystems.

(2) Nanchang arable land was the most widely distributed land, followed by forest land and water. The land type with the greatest change was development land, followed by that of arable land.

(3) ESV continued to decline over the study period, with water and forest land being the major ESV components. Hydrological regulation had the greatest contribution among the individual services; however, maintaining the nutrient cycle had the least

contribution. The high-value areas of Nanchang ecology are mainly located in the northeast corner and in the southeast where water is located. However, the low-value areas are mainly located in the central Nanchang county area.

(4) The Nanchang city CD shows an inverted U-shaped development trend, initially increasing and subsequently decreasing. The CCD also shows an inverted U-shaped trend, initially increasing and subsequently decreasing, with fluctuations in the evolutionary process.

### 5.2. Policy Enlightenment

In the specific implementation process, the main ways to enhance NU can be carried out in the following areas. First, to improve economic urbanization, it is possible to consider improving the degree of development land utilization, concentrating urban development in sub-regions, building industrial clusters, and vigorously developing the economy without taking up ecological land. Meanwhile, we recommend implementing green planning, building green industries, accelerating the conversion of old and new industrial dynamics, accelerating the transformation and upgrading of green structures, comprehensively promoting the green development of Nanchang industries, and vigorously developing circular economy. Social urbanization can be improved by beautifying the living environment, such as building ecological parks and carrying out road greening, and increasing the convenience and comfort of residents' residence, such as building 15-min cities [50,51], in order to improve social urbanization, and also to bring in some talents to settle to a certain extent, thus enhancing population urbanization.

In terms of ESV, the Nanchang has developed the project of "Meiling National Ecological Zone, Mountain and Green", which is expected to continue the improvement of Meiling ESV, and also set up the "three lines and one list" ecological environment zoning control program, that is, the implementation of ecological protection red line, environmental quality bottom line, resource utilization line and ecological environment access list, so as to realize systematic and refined ecological environment management. In implementing the policy, due to the characteristics of unclear suppliers and consumers, incomplete markets and public market externalities, it is necessary to consider strengthening external supervision and clarifying the responsibility and compensation of each link when implementing the policy, and also consider bundling the interests of ecology with the interests of each stakeholder, so that the behavior of protecting the environment can be changed from passive to active.

**Author Contributions:** Conceptualization, Y.H. and Y.L.; methodology, Y.H. and Y.L.; software, Y.H. and Y.L.; validation, Y.H. and Y.L.; formal analysis, Y.H. and Y.L.; investigation, Y.H.; resources, Y.L. and Z.Y.; data curation, Y.L.; writing—original draft preparation, Y.H. and Y.L.; writing—review and editing, Y.H. and Y.L.; visualization, Y.H. and Y.L.; supervision, Y.H., Y.L. and Z.Y.; project administration, Y.H., Y.L., and Z.Y.; funding acquisition, Y.H. All authors have read and agreed to the published version of the manuscript.

**Funding:** This research was funded by the Humanities and Social Sciences Project of Ministry of Education of China, grant number 20YJA630023 and the Science and Technology Project Founded by the Education Department of Jiangxi Province in China, grant number GJJ170997.

**Institutional Review Board Statement:** Not applicable.

**Informed Consent Statement:** Not applicable.

**Data Availability Statement:** The data will be available upon request to the corresponding author.

**Conflicts of Interest:** The authors declare no conflict of interest. The funders had no role in the design of the study; in the collection, analyses, or interpretation of data; in the writing of the manuscript; or in the decision to publish the results.

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
