# Peer review of "Research Regarding the Coupling and Coordination Relationship between New Urbanization and Ecosystem Services in Nanchang"

_sustainability, doi:10.3390/su142215041_

Round 1
Reviewer 1 Report
Research regarding the coupling and coordination relationship between new urbanization and ecosystem services in Nanchang
Introduction: Suggest citing several relevant literatures in the introduction part such as lines 31-34, 36-48, 94-106, and suggest to cite related literature. Overall, the citation is not enough in this part. In addition, somewhere your writing looks like discussions in this section, please revise seriously.
Study area: Could you show a small map of China showing your study area? Tentatively I could not guess your study area in China. If possible keep land use types in your study area map (Figure 1), if not, suggest to keep main cities, highways/railways and main landmarks.
Land use dynamic: What kind of satellite images were used to find out the land use dynamics? Suggest to write its data sources and overall methods.
ESV evaluation method: What is 1/7?
Line 205- Xie et al…… plz write the year.
The study of Xie et al. is a relatively old study, it's about one and half decades old results, still, you are adopting the coefficient values of Xie et al, what do you think is it still applicable to recent studies including your study and other various parts of the Asian countries?
Discussion: Your results show that rapid urbanization resulted in the conversion of arable land into development land in your study area. How about the government's land policies regarding minimizing/ maintaining/ managing rapid urbanization? Please add some government policies, plans, or strategies in your study. Suggest to add more relevant literature in the discussion part to support your study, I think it's not enough.
Reviewer 2 Report
The present study is a high quality work, well empirically supported. It is undoubtedly an interesting topic that has the potential to address the readers of your journal. The structure of the article is logical, with clearly stated objectives and cultivated language. Undoubtedly, these are skilled authors whose erudition is demonstrated by the careful exposition of the results and the detailed description of the research methodology. My comments are focused on the theoretical part of the paper. The conclusions reached by the authors would be better supported by the theoretical background of the study, which should be expanded. It is necessary to mention urbanisation as such - the changes, trends that will continue to have a major impact on urban sprawl and the escalating pressure on land. It is relevant to mention the change in the way of life of urban populations, who are forced, even under the influence of the pandemic, to change their rhythm of life, their habits of relocation before the fear of contagion. Also, the post-pandemic era is constantly bringing new challenges in the research on human commuting behaviour to work or schools to the new geo-social context and its anchoring in space-time. One of the responses to changing the mindset of daily commuting and fulfilling one's health or social needs may be the concept of the 15-minute city, whose central idea is walkability of amenities, workplaces and public spaces from the place of residence. This could reduce the pressure to relocate and expand transport infrastructure and thus reduce the pressure to take agricultural land. See in Mocák P., Matlovičová K., Matlovič R., Pénzes J., Pachura P., Mishra P.K., Kostilníková K, Demková M. (2022). 15-Minute City Concept as a Sustainable Urban Development Alternative: A Brief Outline of Conceptual Frameworks and Slovak Cities as a Case. Folia Geographica 2022, 64/1, pp. 69-89.
Some examples from abroad should have been mentioned in the discussion, for example see Faye Ch. 2019. Organization for the Development of the Senegal River Basin (Omvs) and Integrated Water Resources Management (Iwrm): What Benefits and Difficulties of the Omvs For Iwrm in Senegal? Folia Geographica 2019, 61/1, pp. 17-35, also: Lisnyak A., Utkina K. , Garbuz A. 2018. Present Status of East Forest-Steppe of Ukraine With Reference to Ravine-Beam System of ´Mitrishin Ovrag´. Folia Geographica 2018, 60/1, pp. 62–73. However, the above comments in no way diminish the quality of the study. This is a high-quality study that deals with a very topical issue. It is balanced in content, uses correct methods and I definitely recommend it for publication.
